# Diagnostic Validity in Occlusal Caries Detection of ICDAS II, DIAGNOdent, Radiography and a Combination of the Three Methods: An In Vitro Study

**DOI:** 10.3390/jcm11102937

**Published:** 2022-05-23

**Authors:** Elena Minuesa-García, José Enrique Iranzo-Cortés, Teresa Almerich-Torres, Carlos Bellot-Arcís, José María Montiel-Company, José Manuel Almerich-Silla

**Affiliations:** Departamento de Estomatología, Facultad de Medicina y Odontología, Universidad de Valencia, C/Gascó Oliag Num. 1, 46010 Valencia, Spain; emigar2@alumni.uv.es (E.M.-G.); teresa.almerich@uv.es (T.A.-T.); carlos.bellot@uv.es (C.B.-A.); jose.maria.montiel@uv.es (J.M.M.-C.); jose.m.almerich@uv.es (J.M.A.-S.)

**Keywords:** DIAGNOdent, International Caries Detection and Assessment System (ICDAS), occlusal caries, laser fluorescence, radiography

## Abstract

In this study, we compare the efficacy and diagnostic concordance of the ICDAS, the radiographic criterion and the instrument known as the DIAGNOdent fluorescence laser pen on occlusal caries lesions using a histological section as the gold standard. Of 100 teeth that did not present cavitated occlusal lesions or occlusal fillings, 80 were chosen through a randomization program and examined by two previously trained and calibrated researchers. Subsequently, the teeth were sectioned with a diamond disk and observed under an optical microscope. The results were studied for caries with a limit established in enamel and caries with extension to dentin. The intra-examiner (0.821–0.933) and inter-examiner (0.817–0.924) reproducibility obtained for both ICDAS and DIAGNOdent for the diagnosis of borderline enamel caries was high. Similarly, intra-examiner (0.686–1.000) and inter-examiner (0.809–0.944) reproducibility for diagnosis of caries with dentin extension was also high for both methods. The sensitivity obtained was 0.76 (ICDAS), 0.87 (DIAGNOdent) and 0.58 (Rx), whereas the specificity obtained was 0.66 (ICDAS), 0.4 (DIAGNOdent) and 0.77 (Rx) for lesions limited to enamel. For lesions with extension to dentin, the sensitivity obtained was 0.73 (ICDAS), 0.82 (DIAGNOdent) and 0.09 (Rx), and the specificity obtained was 0.79 (ICDAS), 0.52 (DIAGNOdent) and 0.97 (Rx). Sensitivity increases in both cases by combining diagnostic methods. In conclusion, ICDAS and DIAGNOdent are better diagnostic methods than Rx for the detection of occlusal caries, and the combination of these methods helps to obtain a better diagnosis.

## 1. Introduction

Dental caries is an infectious and multifactorial disease that affects the hard tissues of the tooth. It accounts for—along with the common cold—one of the most prevalent diseases among humans. This term is used to describe the results (signs and symptoms) of a localized chemical dissolution on the tooth surface caused by metabolic processes taking place in the biofilm covering the treated area [1]. In recent decades, the process has become much better defined in several aspects, including microbiology, saliva, tooth mineral composition, tooth ultrastructure, diffusion processes, demineralization kinetics, reversal of demineralization (known as remineralization) and the factors that contribute to the reversal of the process [1]. The disease is initially reversible and can be stopped at any stage of its evolution, even when there is partial destruction of the enamel or dentin (cavitation), as long as the biofilm can be efficiently controlled [2]. The progression of caries in dentin has not only been related to the role of an acidic environment and oral microflora but also to salivary and dentinal metalloproteinases (MMPs) [3].

This concept of the process has caused dentistry professionals to reconsider the diagnostic criteria used to evaluate the presence or absence of disease. Until now, many professionals focused on the disease only as clinically detectable visible lesions to be treated exclusively with surgical treatment; however, currently, it is being diagnosed in much earlier stages with a medical treatment approach aimed at stopping the disease.

Among all the diagnostic methods for caries, the visual and tactile diagnostic method stands out, which consists of examining the tooth surface with the naked eye or with the help of a probe. Among the criteria for visual diagnosis, one of the most used is the International Caries Detection and Assessment System (ICDAS II) criterion. [4]. Exploration following this criterion is carried out using an air syringe to observe, first, the wet surface; then, after drying for 5 s, it is observed again. The parameters that are taken into account to make a diagnosis are the hardness and integrity of the surface and the coloration or color changes. It is a subjective method depending on the experience and training of the examiner, and therefore, it can present a low level of reproducibility in the detection of occlusal caries [5].

Fluorescence is a property of some artificial and natural materials that absorb energy at certain wavelengths and emit light at longer wavelengths. Three different types of fluorescence have been distinguished: blue, with origin in the ultraviolet region of the spectrum; yellow and orange in the ultraviolet region; and red, corresponding to the infrared region of the spectrum [5]. The presence of a carious lesion causes changes in the fluorescent properties of the tooth, which has allowed the development of fluorescence-based methods for the detection and quantification of lesions. These methods are based on the principle that carious dental tissues have altered fluorescence properties compared to healthy dental tissues [6]. This condition is due to the demineralization of the teeth and the metabolic products of bacteria known as porphyrins [7,8].

Most of the fluorescence is induced by the organic components rather than by the disintegration and transmission of the crystals through the inhomogeneous enamel. This hypothesis is based on the fact that laser fluorescence does not detect lesions caused in the laboratory with acids rather than bacterial activity. The bacteria responsible to caries produces certain endogenous porphyrins (fluorophores) that fluoresce when excited by the emission of red laser light [8]. The intensity of fluorescence emission is linearly proportional to the concentration of chromophore molecules (porphyrins) that exist in the target tissue. Porphyrins are excited by visible red light (655 nm), which produces near-infrared emissions. This longer wavelength is used in the KaVo DIAGNOdent system, in which a laser light diode produces a light beam at that wavelength [7]. Said light beam is channeled through the device and is made to affect the tooth, obtaining a bouncing fluorescent light beam that is recorded and quantified, presenting it to the operator as a number. The higher the number, the greater the fluorescence and, by inference, the existence of a more extensive lesion below the surface. This number varies from 0 to 99, with 99 representing the maximum fluorescence [5]. Its most recent version is the Diagnodent Pen^®^, which also enables the diagnosis of proximal lesions (sensitivity of 0.6 and specificity of 1). Due to difficult access to interproximal surfaces, this device features two different tips for each location. The system has shown good performance and reproducibility for the detection and quantification of occlusal and smooth-surface carious lesions in in vitro studies but with somewhat more contradictory results in vivo, both in primary and permanent dentition [5]. Other diagnostic methods based on fluorescence detection have been developed, such as VistaProof, which is based on a camera that takes a photograph and evaluates the reflected fluorescence [9].

Caries is an infectious disease that requires multiple factors for its development. The latest epidemiological studies have shown that the prevalence of caries has been decreasing in recent years, but at the same time, there has been an increase in prevented lesions [10,11,12,13]. This makes early diagnosis of these lesions important so that they can be treated by remineralization and modification of those factors that increase the risk of appearance and development of lesions, thus avoiding cavitation and therefore diminishing the need for conservative treatments for the tooth [9,14,15,16]. For early diagnosis of such lesions, visual inspection is used as a diagnostic method. It is a subjective method based on clinical experience and prior training that has been shown to be safe, accurate and reproducible for the detection of early lesions. Furthermore, in order to quantify lesions more precisely, additional diagnostic methods have been developed, such as DIAGNOdent, which is based on the fluorescence emitted by porphyrins released in caries lesions by cariogenic bacteria [9,15].

The objective of the present study is to evaluate the diagnostic reliability of the visual method, the radiographic method and the DIAGNOdent fluorescence-based system for incipient lesions, as well as to evaluate the possible improvement in diagnosis by combining these systems. The null hypothesis is that combining the methods does not improve the diagnosis of pre-cavitated lesions.

## 2. Materials and Methods

### 2.1. Study Design and Sample Preparation

A sample of 100 teeth recently extracted for periodontal or orthodontic reasons, both healthy and with incipient caries lesions, was collected. Teeth with large cavitated lesions (ICDAS II code 5 or 6) and/or restorations on the occlusal surface were excluded. A KAVO Sonic Flex Lux 2000 L sonic instrument (KaVo, Bilberach/Riss, Germany) was used to clean all teeth and remove any calculus or other debris that might have been present. Subsequently, a contra-angle (MkDent) was used with a brush and prophylaxis paste to finish cleaning and polishing the teeth. Once prepared, the occlusal surfaces of all teeth were photographed. Using a random number generator (https://www.ugr.es/~jsalinas/Aleatorios.htm, accessed on 28 January 2021), 80 of the teeth were selected, and a collaborator who was not involved in the testing process identified the area to be studied using a red circle on each image. All the selected teeth were kept in a physiological saline solution until examination.

### 2.2. Authorizations

This study was approved by the Ethics Committee of the University of Valencia, Spain (registration number 1569195).

### 2.3. Visual Examination Using ICDAS II

Visual examination was performed with a bluish-white spectrum light, always the same device for both explorers, and a triple syringe of air to dry the teeth. On a different sample of teeth, the less experienced explorer who performed the study was calibrated with respect to the second examiner, an expert calibrated in the ICDAS II criteria, obtaining a weighted kappa value of 0.87. Both examiners made a diagnosis of the teeth selected for study.

According to this methodology, the area marked on the dental photographs must first be explored wet and then dried with an air syringe for five seconds to be explored again in order to establish a definitive diagnosis. The following codes were used: Code 0 (healthy); Code 1 (white spot lesion visible after drying the tooth); Code 2 (white spot lesion visible with wet tooth; Code 3 (enamel fracture without exposed dentin); and Code 4 (dark shaded lesion below dentin without complete enamel fracture) [17]. After a week, with the teeth preserved in physiological saline for that time, examination was performed again by both the explorer and the expert examiner to assess intra- and inter-examiner reproducibility.

### 2.4. Examination Using DIAGNOdent

First, a training period took place for the two examiners according to the manufacturer’s instructions. Once this time had elapsed, both examiners explored the selected surfaces of each of the teeth with a DIAGNOdent pen (KaVo, Bilberach/Riss, Germany) to assess inter-examiner reproducibility.

Teeth were first dried with an air syringe kit and calibrated on a healthy tooth surface. Once the zero value (“0”) was established, corresponding to a healthy tooth surface, the tip of the DIAGNOdent pen was placed on the site of the lesion, and the highest value obtained was recorded. This method was repeated for each tooth. For statistical analysis, the values were grouped according to the classification proposed by Lussi et al. in the following four ranges [18]:-0–13: healthy dental surface;-14–20: start of enamel demineralization;-21–29: strong demineralization in enamel;->30: dentin caries lesion.

Both examiners then performed a second scan to assess intra-examiner reproducibility.

### 2.5. Radiological Examination

First, radiographs of the 80 samples were produced using two Dürr brand Vistascan phosphor radiographic plates and a digital developer, shooting from the vestibular surface of the tooth. Subsequently, the two examiners observed the area to be studied once for each of the teeth in the X-rays and established a joint diagnosis based on the following depth codes in radiography [15]:-Code 0: no visible radiolucency;-Code 1–2: radiolucency in the enamel up to the amelodentin limit;-Code 3: radiolucency with fracture of the dentin–enamel line but without obvious progression in the dentin;-Code 4: radiolucency with obvious progression in the outer half of the dentin;-Code 5: radiolucency in the inner half of the dentin.

### 2.6. Histological Analysis

The selected teeth were cut through the area of the lesion marked in the photos with a diamond disc (Komet, Lmgo, Germany), polished with aluminum oxide discs (Sof-lex, 3M ÉSPE) and observed under a microscope (Zeiss, Opmi-Pico, Oberkochen).

The classification proposed by Lussi et al. [18] was applied as follows:-Code 0: caries-free;-Code 1: caries limited to the outer half of the enamel;-Code 2: caries that extends to the inner half of the enamel;-Code 3: caries limited to the outer half of the dentin;-Code 4: caries that extends to the inner half of the dentin.

### 2.7. Data Processing and Statistical Analysis

The data compiled on each form was stored using iOS Numbers spreadsheet software. The identification number of each tooth was recorded, as well as results of the two examinations performed using the ICDAS II criteria by both examiners, those of the two examinations using the DIAGNOdent pen for each of the examiners, the joint result of the radiographic evaluation and the result obtained after histological examination.

To carry out the statistical analysis, a distinction was made between enamel caries and dentin caries. To this end, the codes obtained from the 4 variables (ICDAS, DIAGNOdent, radiography and histology) were first recoded as distinct variables.

Caries with a limit placed in enamel was classified as healthy or decayed as follows:-ICDAS: Code 0 = healthy; codes 1–6 = with a cavity;-DIAGNOdent: Code 0–13 = 0; Codes > 14 = 1;-Rx: Code 0 = 0; Codes 1–5 = 1;-Histology: Code 0 = 0; Codes 1–4 = 1.

Caries extended to dentin was classified as healthy or decayed as follows:
-ICDAS: Codes 0–2 = 0; Codes 3–6 = 1;-DIAGNOdent: Codes 0–29 = 0; Codes > 30 = 1;-Rx: Codes 0–3 = 0; Codes 4 and 5 = 1;-Histology: Codes 0–2 = 0; Codes 3 and 4 = 1;

When the different combinations of diagnostic methods were to be evaluated, the presence of caries was considered as long as at least one of the systems (visual–tactile, radiography or DIAGNOdent) had detected the disease. A second analysis was performed in which the possibility of ruling out the disease was evaluated; this means that the tooth was considered free of caries whenever one of the diagnostic methods ruled out the disease. All calculations of sensitivity, specificity and AUC were based on the first measurements of the most experienced examiner.

Reproducibility was evaluated with the weighted Kappa statistic, following the Landis and Koch classification to assess agreement between examinations as follows [19]:-<0: No agreement;-0.0–0.2: Insignificant;-0.2–0.4: Low;-0.4–0.6: Moderate;-0.6–0.8: Good;-0.8–1.0: Very good.

For the ICDAS sensitivity and specificity analysis, as was the case for the DIAGNOdent analysis and the combination of the different methods, only the first explorations of Examiner 1 were taken into account. Finally, to evaluate the diagnostic reliability of the different systems and histology, the area under the ROC curve was studied.

Statistical analysis was performed using IBM SPSS v.24 software.

## 3. Results

Table 1 shows the inter-examiner reproducibility in examinations 1 and 2 of ICDAS and in examinations 1 and 2 of DIAGNOdent for the detection of borderline caries in enamel (D1) and dentin (D3), as well as the intra-examiner reproducibility.

High intra-examiner and inter-examiner reproducibility is observed. Intra- and inter-examiner reproducibility were similar for ICDAS visual diagnosis, whereas intra-examiner reproducibility was slightly higher in the case of diagnosis with DIAGNOdent.

The following table shows the sensitivity, specificity and area under the curve of the different diagnostic methods, as well as their combinations, with respect to the detection of limits in enamel and dentin (Table 2).

The ROC curves presented in Figure 1a,b correspond to the three investigated methods (ICDAS, DIAGNOdent and radiography) and their combination in lesions limited to enamel and with extension to dentin, respectively.

## 4. Discussion

The sensitivity obtained by DIAGNOdent in this study was 0.87 and 0.82 for the detection of caries in enamel and dentin, respectively. The sensitivity obtained in the present study, compared to that reported in the literature, can be considered high, as several studies obtained a lower sensitivity. Dinitz et al. (2011) [20] obtained a sensitivity of 0.50, and Aktan et al. (2012) [21] obtained a sensitivity of 0.33, whereas the sensitivity obtained in a study by Novaes et al. (2016) was 0.60 [22]. Various studies, such as those by Achileos et al. (2013) [7] and Bussaneli et al. (2015) [23], obtained a slightly higher sensitivity than those previously mentioned but equally lower than ours, i.e., 0.660 and 0.662, respectively. However, in 2008, Rodríguez et al. [24] obtained a sensitivity of 0.78; in 2017, Iranzo-Cortés et al. [14] obtained a sensitivity of 0.85; and in 2015, Ozturk et al. [25] obtained a value of 0.86—all of which are similar to the sensitivity obtained in the present study. The difference in values may be related to the experience of professionals in the use of the devices and the examination protocol. Furthermore, the results could be affected by the type of sample, the cutoff limits used and the solution in which the teeth were stored [21].

For ICDAS, the sensitivity obtained in this study was 0.76 and 0.73 for enamel and dentin, respectively. In 2008, Rodriguez et al. [24] obtained a similar sensitivity (0.73). In other studies, such as those of Diniz et al. (2012) and Shi, Welander and Angmar-Mansson (2000) [26,27], the sensitivity for ICDAS ranged from 0.60 to 0.93. Both methods have high sensitivity, but slightly higher sensitivity can be achieved with the DIAGNOdent diagnostic method [7]. On the other hand, the sensitivity obtained in the present study with the radiographic method was 0.58 and 0.09 for enamel and dentin, respectively, whereas that obtained in a study published by Rodrigues et al. in 2008 [24] was 0.34 and that obtained in a study published by Diniz et al. in 2012 ranged between 0.29 and 0.44 [26]. These results may be due to the fact that occlusal caries with dentin extension is difficult to detect from a buccal surface radiographic view.

The specificity obtained with the DIAGNOdent method was 0.4 and 0.52 for enamel and dentin, respectively, which is lower than that reported in studies, such that by Rodrigues et al. (2008) (0.56) [24] or in a study by Diniz et al. (2012) [26], in which the specificity was 0.89 and 0.85 in teeth with enamel lesions or with lesions extended to dentin, respectively. Côrtes, Ellwood and Ekstrand (2003) [28] obtained a specificity of 0.72 for enamel lesions and 0.91 in the case of lesions that reached the dentin. On the other hand, the specificity obtained with ICDAS in our study was 0.66 and 0.79 for enamel and dentin lesions, respectively, similar to that reported in studies such as that by Rodrigues et al. (200) [24], which was 0.65. In a study published by Diniz et al. (2012) [26] the specificity was 0.60 in lesions limited to enamel and 0.77 in lesions extended to dentin. Regarding radiographic diagnosis, the specificity obtained was 0.77 in lesions limited to enamel and 0.97 in lesions with extension to dentin. A similar specificity (0.97) was obtained in a study by Rodrigues et al. [24], as well as in a study by Diniz et al. [26], in which the specificities obtained were 1.00 for enamel lesions and 0.97 for those that extended to dentin.

As we can see in several studies, including ours, the specificity obtained with the DIAGNOdent system is lower than that obtained with visual diagnosis using ICDAS II, with radiography being the method offering the highest specificity. However, when it comes to sensitivity, it is slightly higher in the case of DIAGNOdent compared to ICDAS and much higher than that of these two methods compared to radiography. The combination of the systems can achieve sensitivity and specificity similar to or greater than those obtained separately.

To determine diagnostic efficacy, the area under the curve was calculated for ICDAS, DIAGNOdent and radiography, as well as the combination of the three methods with respect to histology, which is considered the Gold Standard. The AUC for ICDAS was 0.71 and 0.76, similar to the results reported in other studies, such as that of Diniz et al. (0.86) [26] or higher (0.965), as in Tomczyk et al. [29]. For the DIAGNOdent pen, the area under the curve obtained in the present study was 0.63 and 0.69 for enamel and dentin, respectively, similar to that obtained in studies such as those by Rodrigues et al., Diniz et al. and Onacea et al., who reported AUCs between 0.709 and 0.794 [24,26,30]. On the other hand, the AUC obtained with the radiographic method was 0.69 and 0.52 for enamel and dentin, respectively, being similar or slightly higher than that obtained in other studies, such as those by Rodrigues et al. (0.715) [24], and Diniz et al. (0.65 and 0.74) [26]. We can observe that similar values were obtained with ICDAS and DIAGNOdent, and the radiographic method had slightly lower diagnostic reliability. When combining the diagnostic systems, we observed that the obtained area under the curve was similar to that obtained by each of the methods separately.

The obtained intra-examiner ICDAS reproducibility was 0.871 and 0.821, in lesions limited to enamel, and 1.000 and 0.868 in lesions extended to dentin. In other studies, such as those by Oancea et al., Heinrich-Weltzien et al., Pinelli et al., Attrill et al. or Bakhshandeh et al., similar values were obtained between 0.75 and 0.95 [30,31,32,33,34]. On the other hand, the obtained ICDAS inter-examiner reproducibility was between 0.809 and 0.944, values similar to those obtained in other studies (0.73–0.93) such as those of Hamishaki et al., Heinrich-Weltzien et al., Pinelli et al. or Bakhshandeh et al. [31,32,34,35]. In the present study, high values were obtained, which may be due to the unification of the criteria, as well as training and calibration, as, despite the relative lack of experience of one of the examiners, the obtained values are quite acceptable. On the other hand, inter-examiner reproducibility was slightly higher than intra-examiner reproducibility, and this may be due to the time elapsed between examinations.

Regarding DIAGNOdent, the obtained intra-examiner reproducibility was 0.933 and 0.890 in lesions limited to enamel and 0.923 and 0.919 in lesions with extension to dentin. These results are similar to those reported by other authors, whose values ranged between 0.79 and 0.98 [27,30,31,32,35,36]. On the other hand, the inter-examiner reproducibility obtained with DIAGNOdent (0.862, 0.825, 0.870 and 0.922) was also similar to that reported in other studies (0.77–0.97) such as those of Jablonski-Momeni et al., Pinelli et al. and Rechman et al. [5,32,36].

In the present study, we demonstrated the diagnostic reliability of both the most experienced and the most novice examiner; therefore, we can conclude that the DIAGNOdent can be used by any professional without the need for previous experience, obtaining similar results. Some authors go so far as to state that prior training and calibration does not significantly affect the results obtained by DIAGNOdent [37].

The combination of the investigated methods improves sensitivity, that is, the ability to detect lesions at the D1 level, compared to the use of the different methods individually. This produces a decrease in specificity, resulting in false positives. Because the objective of this study is to evaluate the diagnostic validity of the different methods for the diagnosis of incipient caries, the treatment of which, in most cases, consists of a non-invasive remineralizing regimen for the patient, it could be interesting to investigate the combination of the different diagnostic methods to increase sensitivity. This would lead to a loss of specificity, but because the treatment to be performed on the patient is non-invasive, without harm to the patient in the event that there is no injury, there would be no contraindication to the application of the remineralizing agent.

In the case of dentin lesions (D3), the diagnostic efficiency decreases with the combination of the different diagnostic methods. For this type of lesion, it would be recommended to rely on the result obtained using the ICDAS criterion, as it achieves the greatest area under the ROC curve. In this case, given that the recommended treatment would be surgical, it is important to avoid false positives as much as possible, with the ICDAS being the diagnostic method that has shown greatest validity in this regard.

A possible limitation of the study is the fact that it is an in vitro and not an in vivo study, as in some in vivo studies, the sensitivity obtained with DIAGNOdent is greater than in in vitro studies [38,39]. Nonetheless, other studies indicate that the diagnostic results with DIAGNOdent are somewhat more contradictory in in vivo studies [5], where it is also difficult to justify the histological analysis in teeth diagnosed as healthy. Other possible limitations could be a possible bias in the distribution of the different types of lesions in the sample, as the results were not modified to account for bias, hence remaining completely random.

## 5. Conclusions

After analyzing the results obtained in this study and in other similar studies, we can conclude that:-The combination of the three methods does not significantly improve the diagnostic capacity of occlusal caries lesions, despite showing an improvement over the results of the different methods separately.-A combination of the different methods would be advisable, the radiographic method being the most dispensable, as both the ICDAS II criterion and the DIAGNOdent diagnostic method are more effective independently, obtaining the best results with the combination of the two methods.-The reproducibility of the ICDAS II criteria and the DIAGNOdent system is high for the diagnosis of lesions limited to enamel, as well as those that go deep into dentin.-Both the ICDAS II criterion and the DIAGNOdent system present good results after brief training, similar to those obtained by examiners with more experience.

It would be advisable to carry out new studies that would emphasize an in vivo evaluation of the different diagnostic methods.

## Figures and Tables

**Figure 1 jcm-11-02937-f001:**
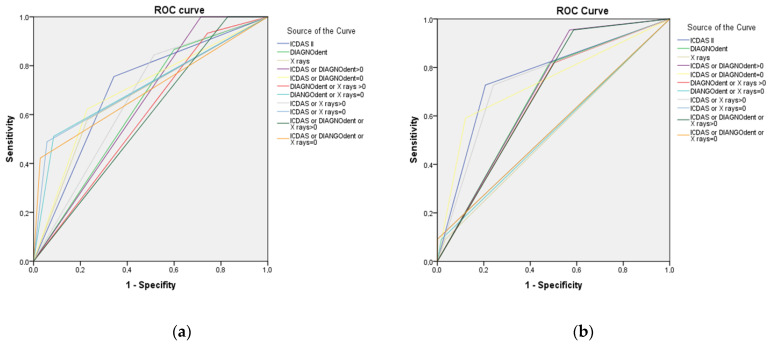
ROC curve for the diagnostic validity of the different methods vs. histology in lesions limited to enamel (**a**) or dentin (**b**).

**Table 1 jcm-11-02937-t001:** Intra- and inter-examiner Reproducibility of ICDAS and DIAGNOdent for lesions limited to enamel (D1) and dentin (D3).

	Inter-Examiner D1	Inter-Examiner D3	Intra-Examiner Examiner 1 D1	Intra-Examiner Examiner 2 D1	Intra-Examiner Examiner 1 D3	Intra-Examiner Examiner 2 D3
**Reproducibility ICDAS first exam (kappa IC 95%)**	0.924 (0.839–1.008)	0.944 (0.868–1.002)	0.871 (0.761–0.980)	0.821 (0.934–0.947)	1.000 (1.000–1.000)	0.868 (0.745–0.979)
**Reproducibility ICDAS second exam (kappa IC 95%)**	0.817 (0.687–0.946)	0.809 (0.675–0.944)				
**Reproducibility DIAGNOdent first exam (kappa IC 95%)**	0.862 (0.731–0.993)	0.870 (0.760–0.979)	0.933 (0.842–1.025)	0.890 (0.769–1.012)	0.923 (0.838–1.008)	0.919 (0.830–1.009)
**Reproducibility DIAGNOdent second exam (kappa IC 95%)**	0.825 (0.677–0.972)	0.922 (0.835–1.008)				

**Table 2 jcm-11-02937-t002:** Sensitivity, specificity and area under the ROC curve for ICDAS, DIAGNOdent, radiography and the combination of the three methods for the detection of caries lesions limited to enamel (D1) or dentin (D3).

	Sensitivity D1	Specificity D1	Area under the ROC Curve D1	Sensitivity D3	Specificity D3	Area under the ROC Curve D3
**ICDAS II**	0.76	0.66	0.71 (0.59–0.82)	0.73	0.79	0.76 (0.64–0.88)
**DIAGNOdent**	0.87	0.4	0.63 (0.51–0.76)	0.82	0.52	0.69 (0.54–0.80)
**Radiography**	0.58	0.77	0.68 (0.56–0.79)	0.09	0.97	0.53 (0.39–0.65)
**DIAGNOdent or ICDAS > 0**	1	0.29	0.64 (0.52–0.77)	0.95	0.43	0.69 (0.56–0.81)
**DIAGNOdent or ICDAS = 0**	0.62	0.77	0.70 (0.58–0.81)	0.59	0.88	0.74 (0.60–0.87)
**DIAGNOdent or Rx > 0**	0.93	0.26	0.56 (0.47–0.72)	0.82	0.5	0.66 (0.53–0.79)
**DIAGNOdent or Rx = 0**	0.51	0.91	0.71 (0.60–0.83)	0.09	0.98	0.54 (0.39–0.68)
**ICDAS or Rx > 0**	0.84	0.49	0.67 (0.54–0.79)	0.73	0.76	0.74 (0.62–0.87)
**ICDAS or Rx = 0**	0.49	0.94	0.72 (0.60–0.83	0.09	1	0.55 (0.40–0.69)
**ICDAS or DIAGNOdent or Rx > 0**	1	0.17	0.59 (0.46–0.71)	0.95	0.41	0.68 (0.57–0.80)
**ICDAS or DIAGNOdent or Rx = 0**	0.42	0.97	0.697 (0.58–0.81)	0.09	1	0.54 (0.40–0.69)

## Data Availability

Not applicable.

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
