# Peer review of "Diagnostic Validity in Occlusal Caries Detection of ICDAS II, DIAGNOdent, Radiography and a Combination of the Three Methods: An In Vitro Study"

_jcm, 2022, doi:10.3390/jcm11102937_

Round 1

Reviewer 1 Report

Dear authors, 

I find your article very interesting but there are a few minor changes that need to be made. 

Please introduce in introduction more informations about the equipment used in detecting the caries.

Please introduce the null-hypotheses.

Line 118 please correct - one sample of 100 teeth, please change the explanation in order to be easy to undestand how much samples were included in the manuscript.

Conclusion  I suggest to rewrite three major conclusions and not a long paragraph.

Author Response

Dear reviewer,

Thank you very much for your revision and comments. We will try to answer every comment you made.

Please introduce in introduction more informations about the equipment used in detecting the caries.

We have added more information about other diagnostic methods.

Please introduce the null-hypotheses.

We have added it with the objective of the study

Line 118 please correct - one sample of 100 teeth, please change the explanation in order to be easy to undestand how much samples were included in the manuscript.

We have changed it.

Conclusion  I suggest to rewrite three major conclusions and not a long paragraph.

We have changed it

Reviewer 2 Report

Dear Authors

the manuscript is interesting and suitable for the newspaper although some lacks in method and design have emerged

INTRODUCTION

Please provide an explicit statement of the objectives or questions the paper addresses.

  • “one of the most 60 used is the ICDAS II criterion”: please the first time, do not enter the contract name but enter it in full

The MATERIALS AND METHODS section presents some lacks:

  • “Teeth with large cavitated lesions and/or restorations on the occlusal surface were excluded”:

Please define the criteria to define a large cavity

  • “a random number generator”: please define the method to obtain a random number generator
  • “the occlusal surfaces of all teeth were photographed “: please define the method to obtain a random number generator
  • “Both examiners made a diagnosis of the teeth selected 136 for study”: please explain how the examiners were selected. more information regarding their selection criteria should be added
  • “Teeth were first dried with an air syringe kit and calibrated on a healthy tooth surface. Once the zero value ("0") was established”: please explain if this evaluation was performed on the same teeth as the previous one and how many times later than the first one.
  • “Radiological examination”: please clarify how many times later these measurements were performed.

DISCUSSION

The discussion is too little developed, it should be implemented with some considerations and reflections on what emerged from the analyzed data

REFERENCES

Please consider to add these reference to yours

Femiano F, Femiano R, Femiano L, Jamilian A, Rullo R, Perillo L. Dentin caries progression and the role of metalloproteinases: an update. Eur J Paediatr Dent. 2016 Sep;17(3):243-247.  

Author Response

Dear reviewer,

Thank you very much for your revision and comments. We will try to answer every comment you made.

INTRODUCTION

Please provide an explicit statement of the objectives or questions the paper addresses.

  • “one of the most 60 used is the ICDAS II criterion”: please the first time, do not enter the contract name but enter it in full

 We have added the full name in the review.

The MATERIALS AND METHODS section presents some lacks:

  • “Teeth with large cavitated lesions and/or restorations on the occlusal surface were excluded”: Please define the criteria to define a large cavity

We have defined it in the review.

  • “a random number generator”: please define the method to obtain a random number generator

We have defined it in the review.

  • “the occlusal surfaces of all teeth were photographed “: please define the method to obtain a random number generator

We have defined it in the review.

  • “Both examiners made a diagnosis of the teeth selected for study”: please explain how the examiners were selected. more information regarding their selection criteria should be added

One of the examiners was a calibrated examiner in ICDAS and the other examiner was a post-graduated student.

  •  “Teeth were first dried with an air syringe kit and calibrated on a healthy tooth surface. Once the zero value ("0") was established”: please explain if this evaluation was performed on the same teeth as the previous one and how many times later than the first one.

This method was repeated for each tooth. We have defined it in the review.

  • “Radiological examination”: please clarify how many times later these measurements were performed.

We have defined that the radiological examination was performed once.

DISCUSSION

The discussion is too little developed, it should be implemented with some considerations and reflections on what emerged from the analyzed data

REFERENCES

Please consider to add these reference to yours

Femiano F, Femiano R, Femiano L, Jamilian A, Rullo R, Perillo L. Dentin caries progression and the role of metalloproteinases: an update. Eur J Paediatr Dent. 2016 Sep;17(3):243-247.  

The discussion is focused to the comparison between the diagnosis obtained by the different diagnostic methods:  the visual exam, X-ray and laser methods. We consider that our discussion includes the comparison to the most relevant publications in the field. The objective of our study is not to discuss the ethiology of the progression of caries lesions or the role of metalloproteinases.

Round 2

Reviewer 1 Report

dear authors, 

 your manuscript improved and therefore I agree to publication.

Reviewer 2 Report

Dear Authors,

Thank you to review your paper. many improvements were performed in the manuscript.